# Monolithic Integration of Semi-Transparent and Flexible Integrated Image Sensor Array with a-IGZO Thin-Film Transistors (TFTs) and p-i-n Hydrogenated Amorphous Silicon Photodiodes

**DOI:** 10.3390/nano13212886

**Published:** 2023-10-31

**Authors:** Donghyeong Choi, Ji-Woo Seo, Jongwon Yoon, Seung Min Yu, Jung-Dae Kwon, Seoung-Ki Lee, Yonghun Kim

**Affiliations:** 1Department of Energy and Electronic Materials, Surface Nano Materials Division, Korea Institute of Materials Science (KIMS), Changwon 51508, Republic of Korea; baru2300@kims.re.kr (D.C.); wldn0816@kims.re.kr (J.-W.S.); jwyoon@kims.re.kr (J.Y.); jdkwon@kims.re.kr (J.-D.K.); 2School of Materials Science and Engineering, Pusan National University, Busan 46241, Republic of Korea; 3Analytical Research Division, Korea Basic Science Institute, Jeonju 54907, Republic of Korea; smyu0409@kbsi.re.kr

**Keywords:** oxide TFT, a-IGZO, a-Si:H, photodiode, image sensor, flexible, transparent, low-temperature process

## Abstract

A novel approach to fabricating a transparent and flexible one-transistor–one-diode (1T-1D) image sensor array on a flexible colorless polyimide (CPI) film substrate is successfully demonstrated with laser lift-off (LLO) techniques. Leveraging transparent indium tin oxide (ITO) electrodes and amorphous indium gallium zinc oxide (a-IGZO) channel-based thin-film transistor (TFT) backplanes, vertically stacked p-i-n hydrogenated amorphous silicon (a-Si:H) photodiodes (PDs) utilizing a low-temperature (<90 °C) deposition process are integrated with a densely packed 14 × 14 pixel array. The low-temperature-processed a-Si:H photodiodes show reasonable performance with responsivity and detectivity for 31.43 mA/W and 3.0 × 10^10^ Jones (biased at −1 V) at a wavelength of 470 nm, respectively. The good mechanical durability and robustness of the flexible image sensor arrays enable them to be attached to a curved surface with bending radii of 20, 15, 10, and 5 mm and 1000 bending cycles, respectively. These studies show the significant promise of utilizing highly flexible and rollable active-matrix technology for the purpose of dynamically sensing optical signals in spatial applications.

## 1. Introduction

The field of transparent and high-performance thin-film transistors (TFTs) has witnessed remarkable advancements, with amorphous indium–gallium-zinc oxide (a-IGZO) thin-film transistors emerging as a promising material [1,2,3,4,5,6,7]. These TFTs exhibit exceptional field-effect mobility, along with impressive on/off switching exceeding 10^8^ [8,9]. Such attributes have positioned them as compelling candidates for cutting-edge applications in flat-panel displays and imagers [10,11,12].

Notably, the integration of a-IGZO TFTs with organic light-emitting diodes (OLEDs) has facilitated the development of transparent and efficient display technologies [13,14,15,16]. Furthermore, coupling a-IGZO TFTs with perovskite materials has paved the way for high-performance image sensors [17,18,19]. Also, there is a current surge in research focused on the advancement of photodiodes utilizing organic materials [20]. This combination underscores the versatility and potential of a-IGZO TFTs in diverse transparent and flexible optoelectronic applications. In contrast to conventional amorphous silicon TFTs, oxide TFTs offer significant advantages for the development of high-performance image sensors. They boast superior on/off ratios and frame rates, enabling the realization of advanced image-capturing capabilities [21]. Furthermore, the use of transparent conductive oxide (TCO) materials like indium tin oxide (ITO) for both gates and bottom-top electrodes has enabled the fabrication of all-transparent TFTs [22,23]. This innovation not only ensures low voltage and power consumption but also enhances energy efficiency, rendering these devices particularly attractive for integration into smart devices, in-car displays, and wearable electronics [24,25,26,27]. Also, a blue light detector can be typically used for blue light hazards [28]. It prominently functions at 470 nm, making it suitable for use as an image sensor. This capability positions it for applications in areas such as cameras, where imaging sensing is required [29]. To achieve a transparent image sensor and enhanced flexibility, the utilization of a colorless polyimide (CPI) substrate has been employed, effectively minimizing the curvature radius [30]. This strategic implementation addresses the need for improved flexibility while ensuring optimal performance in the context of the advanced optical materials domain [31]. However, most transparent and flexible image sensor arrays are limited to photolithographic patterning and the large spatial uniformity of photodiodes, such as perovskite and organic-based photosensing materials. In this regard, hydrogenated amorphous silicon-based (a-Si:H) photodiodes (PDs) exhibit a remarkable combination of exceptional performance and cost-effectiveness [32]. Their high efficiency, coupled with reasonable production costs, has contributed to their recognition in various optoelectronic applications. These a-Si:H photodiodes are characterized by their heightened sensitivity within the visible spectrum and rapid response times, making them well-suited for capturing light in real-time scenarios [33].

In this study, a transparent and flexible 1T-1D image sensor array on a flexible CPI substrate is successfully implemented using laser lift-off (LLO) techniques. By utilizing transparent ITO electrodes and a-IGZO channel-based TFT backplanes, vertically aligned p-i-n a-Si:H photodiodes are monolithically integrated. These photodiodes are fabricated using a low-temperature (<90 °C) deposition process using the PECVD method and are densely packed in a 14 × 14-pixel configuration. Notably, the low-temperature-processed a-Si:H photodiodes exhibit photo-responsible performance metrics, demonstrating a responsivity (R) of 31.43 mA/W and a detectivity (D*) of 3.0 × 10^10^ Jones (biased at −1 V) at a wavelength of 470 nm. Furthermore, the flexible image sensor arrays exhibit exceptional mechanical durability. They can be attached to curved surfaces with bending radii from 5 to 20 mm, enduring up to 1000 cycles stably. These findings show considerable promise for the integration of flexible active-matrix technology, particularly for real-time optical signal detection in spatial applications.

## 2. Materials and Methods

Synthesis of CPI film and LLO process: The glass substrate was cleaned via sonication in acetone, isopropyl alcohol (IPA), and deionized water (D.I. water) for 5 min. After cleaning, the glass was dried in an oven at 100 °C for 5 min. Subsequently, the cleaned glass substrate was loaded into a plasma-enhanced chemical vapor deposition chamber (PECVD, SNTEK, Anseong, Republic of Korea), and an a-Si:H sacrificial layer with a thickness of approximately 100 nm was deposited after generating plasma with very high-frequency (VHF) 30 W power under H_2_ and SiH_4_ conditions at 250 °C and 400 mTorr. To form the CPI film, we prepared a solution by stirring IPI-C (5000 cps, solid content = 18.5 wt%, IPITECH, Daejeon, Republic of Korea) and 3-glycidoxypropyltrimethoxysilane (98%, Sigma-Aldrich, St. Louis, MO, USA) together. The solution was then spin-coated onto the a-Si:H-deposited glass substrate at a rotation speed of 400 rpm. The solution-coated glass substrate was annealed at 350 °C for 4 h in a box furnace (Thermal System and Technology, Hwaseong, Republic of Korea). The a-Si:H sacrificial layer sandwiched between the glass substrate and the CPI film was removed using a laser with a wavelength of 532 nm. The laser was controlled using an external attenuator, frequency, and diode current (Kortherm Science, Incheon, Republic of Korea).

Fabrication of a-IGZO TFT backplane: The surface roughness of the CPI film after the LLO process was reduced by depositing a 50 nm thick Al_2_O_3_ buffer layer using atomic layer deposition (ALD, Lucida M 100, NCD, Seoul, Republic of Korea) [34]. The deposition was carried out at 200 °C utilizing a trimethylaluminium ((CH_3_)_3_Al, TMA) precursor and H_2_O [35]. The 80 nm thick ITO gate electrode was fabricated via direct current (DC) magnetron sputter at room temperature (R.T) with Ar flow at 25 sccm and O_2_ flow at 0.18 sccm under 3.50 mTorr of pressure. After deposition, the ITO gate electrode was defined by a photolithography process followed by the selective wet etching of the ITO using copper etchant solution (Sigma-Aldrich, USA). Following that, a 50 nm Al_2_O_3_ gate dielectric layer was deposited using the ALD system with the TMA precursor and H_2_O at 200 °C. For the 40 nm thick a-IGZO (In_2_O_3_:Ga_2_O_3_:ZnO = 1:1:2, TASCO, Anyang, Republic of Korea) channel, 3-inch radio frequency (RF) sputter (Ultech, Daegu, Republic of Korea) was employed at R.T. with Ar flow at 25 sccm and RF power set at 80 W under 8 mTorr of pressure. After deposition, a photolithography process was carried out for channel patterning, followed by the selective wet etching of a-IGZO using 1 M hydrochloric acid solution (HCl, Samchun Chemical Co., Seoul, Republic of Korea). Subsequently, S/D patterning was patterned the utilization of the photolithography process. Following this, ITO source and drain (S/D) electrode deposition was conducted using a DC magnetron sputtering technique at room temperature, maintaining a working pressure of 3.50 mTorr. The deposition process was carried out with a controlled Ar flow rate of 25 sccm and an O_2_ flow rate of 0.18 sccm. The lift-off process used acetone sonication. Finally, a 50 nm thick Al_2_O_3_ passivation layer was deposited using the ALD system with the TMA precursor and H_2_O at 200 °C.

Fabrication of p-i-n a-Si:H PD: First, to open the bottom ITO electrodes connected to the drain electrodes of the a-IGZO TFTs, photolithography patterning was employed. Subsequently, phosphoric acid (H_3_PO_4_ Samchun Chemical Co., Republic of Korea) etching was conducted at 55 °C for 40 s to etch away 50 nm of the Al_2_O_3_ passivation layer. Afterward, to prevent damage to the ITO electrodes from plasma exposure, a 30 nm thick layer of aluminum zinc oxide (AZO) was deposited using DC magnetron sputtering. Next, three layers of n-Si (20 nm), i-Si (400 nm), and p-Si (30 nm) were deposited using PECVD. Finally, the ITO top electrode was deposited to a thickness of 200 nm using DC magnetron sputtering at room temperature with Ar flow at 20 sccm and O_2_ flow at 0.18 sccm under 3.50 mTorr of pressure.

Characterization: The elemental composition of a-IGZO was analyzed utilizing XPS (NEXSA, Thermo Fisher Scientific, Waltham, MA, USA). The cross-sectional and elemental analyses of both the a-IGZO TFT and the a-Si:H photodiode were conducted via SEM imaging and EDS mapping. The electrical performance assessment was carried out using the Keithley 4200-SCS semiconductor analyzer system at room temperature and under atmospheric pressure conditions. For light measurements, a controlled environment was maintained within a dark shield box (MSD1, MS tech), where an LED controller (BioLED light source control module, Mightex, Toronto, ON, Canada) and a 470 nm light source (BLS-LCS-0470-03-22, Mightex) were connected. Furthermore, the flexibility of the image sensor was evaluated by performing bending tests using a linear stage (M-433, Newport, Irvine, CA, USA).

## 3. Result and Discussion

### 3.1. Device Structure and Fabrication

Figure 1a illustrates the overall 3D scheme of an a-IGZO TFT and an a-Si:H photodiode (1T-1D) image sensor. The monolithic integrated image sensor consists of an a-IGZO thin-film transistor for sensing and switching and an a-Si:H p-i-n photodiode for photo conversion. In this device configuration, the role of the a-IGZO TFT is to regulate its on/off states through the gate field, effectively managing current flow because of its high conductivity and rapid response characteristics. Furthermore, the p-i-n a-Si:H photodiode transforms incoming light into electrical signals via the photoelectric effect. Incident light generates charge carriers (electron–hole pairs) in the i-layer, converting light energy into electrical signals. Integrating a photodiode within the 1T-1D structure enables light-induced electron–hole pair generation. Figure 1b shows a scanning electron microscopy (SEM) image of the 1T-1D configuration. Also, an optical microscopy (OM) image of the monolithic integration of the image sensor array is depicted in Figure 1c. Figure 1d shows an atomic force microscope (AFM) image of the a-IGZO surface. An AFM analysis was conducted in a selected area of 1000 nm × 1000 nm, and the root mean square (RMS) roughness was approximately 0.06 nm. This result shows that the surface of the a-IGZO has excellent uniformity [36,37].

Figure 1e presents the overall fabrication process for the vertically stacked 1T-1D image sensor array. In order to fabricate the flexible 1T-1D image sensor array on a flexible substrate, LLO was used to release the device on the rigid glass substrate [34,38]. Initially, a CPI film was uniformly coated with a thickness of 20 um on a glass substrate with an area of 5 × 5 cm^2^. Subsequently, a-IGZO TFTs with bottom local-gate structures were manufactured using the conventional photolithography technique. To maintain the optical transparency of our flexible TFTs, the ITO electrodes were used to form the gate, source, and drain electrodes. Then, a p-i-n a-Si:H was deposited using PECVD at a temperature of 90 °C. Finally, the CPI film was peeled off from the glass substrate using the LLO process. A comprehensive schematic of the intricate component fabrication process is illustrated in Appendix A. Furthermore, we present optical microscopy (OM) images captured of each fabrication step in Appendix A. Additionally, the transmittance spectra of the IGZO film deposited on the glass substrate are shown in Appendix A.

SEM and energy-dispersive X-ray spectroscopy (EDS) maps are also shown to analyze the distribution of elements within the a-IGZO and a-Si:H photodiode, confirming the homogeneous distribution of the elements, as shown in Figure 2a,b. Additionally, a surface analysis of the a-IGZO channel material using X-ray photoelectron spectroscopy is shown in Appendix A. At the In 3d peak, the binding energy values for In 3d^5/2^ and In 3d^3/2^ were observed at 444.1 and 452 eV, respectively. These are attributable to the In^3+^ binding state in the IGZO. At the Ga 2p peak, the binding energy values for Ga 2p^3/2^ and Ga 2p^1/2^ were found to be 1117.3 and 1144.1 eV, respectively, indicating Ga^3+^ binding states in the IGZO. At the Zn 2p peak, the binding energy values for Zn 2p^3/2^ and Zn 2p^1/2^ were determined as 1021.3 and 1044.4 eV, respectively, suggesting Zn^2+^ binding states in the IGZO. The O 1s peak of the IGZO separates into two peaks at 529.8 and 531 eV. The lower binding energy peak (529.8 eV) corresponds to the binding energy of the lattice oxygen (metal-oxide, O_I_), while the peak at 531 eV is associated with non-lattice oxygen (oxygen vacancy, O_II_). The areas obtained from the deconvolution of the O 1s peak (A_OI_ and A_OII_) are utilized for the quantitative analysis of oxygen vacancies in the IGZO film using the following formulae [39,40]: (1)$$O_{I}\;(\%)=\frac{\mathrm{AO}I}{\mathrm{AO}I+\mathrm{OII}}\times 100$$
(2)$$O_{I}\;(\%)=\frac{\mathrm{AOII}}{\mathrm{AO}I+\mathrm{OII}}\times 100$$

Equations (1) and (2) are used to calculate the O_I_ and O_II_ area ratios of IGZO. The extracted IGZO O_I_ and O_II_ area ratios are 60.90% and 39.10%, respectively.

### 3.2. Electrical Characteristics of a-IGZO TFT Backplanes

The vertical scheme and electrical measurement configuration for the 1T-1D device cell are shown in Figure 3a. Figure 3b shows the transfer characteristics (I_d_-V_g_) measured with 196 individual a-IGZO TFTs. The overall 196 a-IGZO TFTs fabricated on CPI flexible film show a device yield of 100 %, and the device could work well with tight distribution and low variability. This means that uniform switching functionality connected to a-Si photodiodes is possible. The output characteristics (I_d_-V_d_) were investigated while varying the gate voltage (V_g_) from 0 to 15 V in increments of 1 V, as shown in Figure 3c. This investigation revealed a consistent increase in drain current (I_d_) as V_g_ increased.

Figure 3d,e present statistical histograms depicting the device parameters for the on/off ratio and field-effect mobility, respectively. The average value of the on/off ratio (I_on_/I_off_) was found to be about 10^8^. The field-effect mobility (μ_FE_) was determined with an average value of 2.24 cm^2^/V·s and a standard deviation of 1.43, calculated using the following formula:(3)$$\mu_{\mathrm{FE}}=\left(\frac{dI_{D}}{dV_{G}}\right)\left(\frac{L}{V_{D}C_{ox}W}\right)$$

In this equation, *I_D_*, *V_G_*, *L*, *V_D_*, *C_ox_*, and *W* represent the source-drain current, the gate voltage, the channel length, the drain voltage, the gate capacitance of the Al_2_O_3_ dielectrics, and the channel width, respectively.

Figure 3f illustrates the statistical distribution of threshold voltage (V_th_) for a total of 196 individual a-IGZO TFTs. The average value and standard deviation are estimated to be approximately 6.45 and 1.43 V, respectively. As a result, the fabricated a-IGZO TFT array shows good uniformity and reproducibility. A comparison of the electrical characteristics between the a-IGZO-isolated TFTs and those of the TFT after photodiode integration is depicted in Appendix A. It is confirmed that there is a negligible change in the device parameters for the on/off ratio and the field-effect mobility.

### 3.3. Photoresponsivity Characteristics

The images of the measurements in dark and photo conditions are presented in Appendix A. Dark current measurements were performed in a dark shield box to block all external light. For photocurrent measurements, the device was positioned at a 5 cm distance from the light source within the dark shield box, and the light intensity was controlled under a wavelength of 470 nm light illumination.

Figure 4a,b illustrate the J-V characteristics of a p-i-n a-Si:H photodiode and the transfer characteristics upon the integration of an a-IGZO TFT with a p-i-n a-Si:H photodiode. It was found that the photocurrent also increased as the light intensity increased. Figure 4c shows the photocurrent behavior as an increase in light intensity, directly correlating with a proportional rise in photocurrent. The p-i-n a-Si:H photodiode operates on the principle of generating electron–hole pairs upon light illumination. Increasing light intensity induces a proportional increase in the creation of electron–hole pairs. Subsequently, these generated electron–hole pairs facilitate the generation of electric current as they traverse the material, thereby leading to an augmented photocurrent output. The photocurrent exhibited an almost linear correlation with the increasing voltage, suggesting the formation of ohmic contact between the top and bottom and top ITO electrodes within the a-Si:H photodiode structure. At a voltage bias of 1 V, the photocurrent (*I_ph_*), characterized as a function of incident light intensity, was evaluated utilizing a simplified power law equation, as depicted in Figure 4c. In this equation, *A* signifies the wavelength constant of the incoming light. Additionally, the power law equation is calculated using the following formula:(4)$$I_{ph}=A\times P^{\alpha }$$

In the power law equation, *P* represents the applied light intensity, and α corresponds to the exponent denoting the degree of photosensitivity [41]. The experimental data exhibited a favorable fit with the power law equation, exhibiting a linearity of 0.94 [42]. Figure 4d,e illustrate that the important parameters in the evaluation of photodiode effectiveness are the dependence of responsivity (R) and specific detectivity (D*) on incident light intensity. These two parameters can be obtained from the following formulas:(5)$$R=\frac{I_{ph}}{P_{opt}\cdot A}$$
(6)$$D^{*}=R\sqrt{\frac{AB}{\bar{l_{n}^{2}}}}$$

In this equation, *I_ph_*, *P_opt_*, *A*, *B*, and $\bar{l_{n}^{2}}$ are the photocurrent, light intensity, effective area, frequency bandwidth, and mean-square noise current measured at a bandwidth of 100 Hz in darkness, respectively [43]. Figure 4d,e illustrate the R and D* responses at −1 V for various light intensities. The maximum responsivity of 31.43 mA/W was accompanied by a specific detectivity value of 3.00 × 10^10^ Jones (1 Jones = 1 cm Hz^1/2^/W). Responsivity quantifies the efficiency of light-to-electrical-signal conversion, whereas detectivity evaluates the device’s ability to discern weak signals despite background noise. The measurement plots of the noise current within a frequency range of 1 Hz to 10 kHz in an integrated image sensor containing a-IGZO TFT and a-Si:H photodiodes are presented in Appendix A.

The dynamic photoresponse of the p-i-n a-Si:H photodiode was observed as depicted in Figure 4f. This response was observed under a cycle of on/off lasers (470 nm) at intervals of 0.3 s, repeating in cycles of 1000. The drain current increased during illumination and decreased upon the deactivation of the laser. The rise time (τ_r_) associated with on/off light is approximately 140 ms, while the fall time (τ_f_) is 21 ms, as shown in Appendix A. This photodynamic response persists for 500 s over 1000 on/off cycles, showcasing repetitive and stable light reactions [44].

Figure 5 shows the photocurrent mapping using an active pixel image sensor of a 14 × 14 array, showing “K”, “I”, “M”, and “S”. The number of devices corresponding to each character is 76 for K and I, 116 for M, and 98 for S. Except for the parts corresponding to the characters in the four characters, the shadow mask was covered and measured at V_DS_ = 1 V and V_GS_ = 15 V. The part receiving external light illumination is expressed as I_ph_. Accordingly, these results show the excellent operational performance of an integrated image sensor that combines a-IGZO TFT and p-i-n a-Si:H photodiodes, demonstrating remarkable electrical and optical uniformity [45].

### 3.4. Structural Flexibility

Flexibility and rollability are crucial mechanical attributes that play a significant role in the operation of flexible image sensors [42]. An integrated image sensor has been developed, featuring a combination of a-IGZO TFTs and a-Si:H photodiodes fabricated on a flexible CPI film. Remarkably, this integrated sensor maintains its performance integrity even when subjected to rigorous bending tests and high bending radii, as detailed in Table 1. Figure 6a presents images from a 1T-1D image sensor in a bending state with a diameter of 20 mm. Photoelectric measurements were conducted for various diameters, ranging from 10 to 40 mm. Figure 6b depicts the photo and dark states measured based on the transfer characteristics (V_d_ = 1 V, V_g_ = 15 V) with various bending radii: flat, 5, 10, 15, and 20 mm. Few differences in the dark current and photocurrent were found when the bending radii (r) varied at 5 mm, 10 mm, 15 mm, and 20 mm. Figure 6c demonstrates the dynamic response when applying a light pulse with a 0.3 s duration for approximately 600 s at V_d_ = 1 V while the bending radius was set at 5 mm. The rise time is approximately 200 ms, and the fall time is around 300 ms. The photodynamic response exhibits a repeated stable photoresponse for 600 s, indicating that few current variations could be observed.

In Figure 6d, images from a flexible image sensor integrated with a 1T-1D structure under upward bending are shown. Maximum upward bending is applied with a bending radius of 5 mm in 200 bending cycles, ranging from 200 to 1000 cycles. Figure 6e illustrates the corresponding photo and dark states measured based on the transfer characteristics (V_d_ = 1 V, V_g_ = 15 V) with respect to the bending cycles. This reveals that the difference in current between the dark and photocurrent states remains low despite the bending cycles. Figure 6f displays light pulse measurements of the image sensor taken during a 1000-cycle upward bending test, where V_d_ is set at 1 V for approximately 600 s. Similar to previous observations, τ_r_ is about 200 ms, τ_f_ is around 300 ms, and there is little current difference.

In Figure 6g, images acquired from the flexible image sensor integrated with a 1T-1D structure under downward bending are displayed. Similar to the upward bending test, measurements were conducted with maximum incremental downward bending with a bending radius of 5 mm over 200 bending cycles, ranging from 200 to 1000 cycles. Figure 6h depicts the corresponding and transfer characteristics (V_d_ = 1 V, V_g_ = 15 V) based on the bending cycles. The difference in current change between the dark and photo states remains negligible. Furthermore, Figure 6i showcases light pulse measurements of the image sensor during a 1000-cycle downward bending test, with V_d_ set at 1 V for approximately 600 s. These results show its remarkable mechanical and physical durability. 

Additionally, the deformation ratio exerted on each device according to its diameter was calculated, as described below:(7)$$\varepsilon_{TFT}\;\left(\%\right)=(t_{CPI}+t_{Al_{2}O_{3}}+t_{ITO}+t_{a-IGZO})\;/\emptyset \times 100$$
(8)$$\varepsilon_{Photodiode}\;\left(\%\right)=(t_{CPI}+t_{Al2O3}+t_{ITO}+t_{a-Si:H})\;/\emptyset \times 100$$

In this equation, *ε*, *t*, and ∅, are the strain, thickness of the layer, and diameter of the device [34]. The stable and reproducible endurance characteristics remain even after undergoing 1000 cycles, even when the device is folded to a diameter of 5 mm (ε_tft_ = 0.20%/ε_photodiode_ = 0.21%). A comparison of the dark current and photocurrent, as indicated by the J-V characteristics measured under different bending radii and bending cycles, is illustrated in Appendix A.

## 4. Conclusions

In this study, we successfully synthesized p-i-n a-Si:H photodiodes at low process temperatures (<90 °C) and integrated them with a-IGZO TFTs to achieve a transparent 1T-1D-structured image sensor array. The implementation of this structure on a flexible ultrathin CPI film substrate exhibited remarkable on/off ratios and field-effect mobility. Furthermore, under a 470 nm wavelength, the image sensor demonstrated excellent responsivity exceeding 31.43 mA/W, along with exceptional uniformity and yield. Remarkably, even after undergoing more than 500 bending tests, the device maintained superior flexibility and mechanical stability.

In conclusion, our research establishes the feasibility of a transparent 1T-1D image sensor array composed of p-i-n a-Si:H photodiodes and a-IGZO TFTs, synthesized at low process temperatures. The exceptional performance, including impressive responsivity, uniformity, and mechanical stability, positions this integrated structure as a promising candidate for a wide range of flexible and transparent electronic applications. Our findings underscore the potential of this technology to drive advancements in the field of flexible electronics and pave the way for innovative applications in various industries.

## Figures and Tables

**Figure 1 nanomaterials-13-02886-f001:**
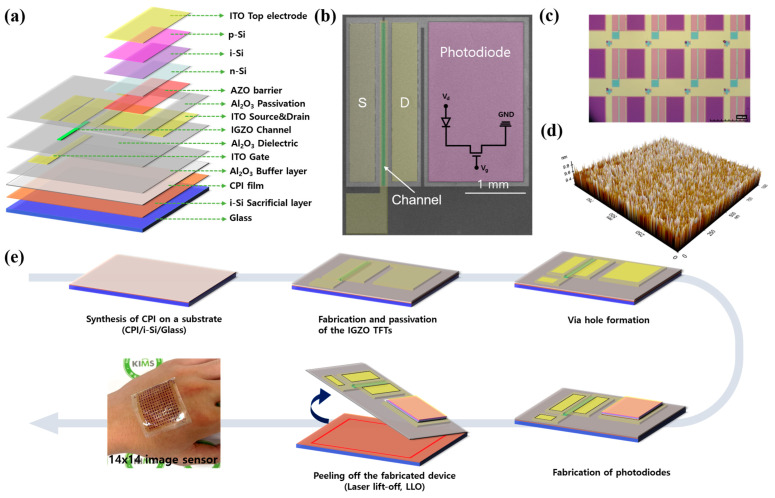
(**a**) Schematic of the a-IGZO TFT and a-Si:H photodiode. (**b**) SEM image of a-IGZO TFT/a-Si:H photodiode. (**c**) Optical microscopy (OM) image of the monolithic integration of the image sensor array. (**d**) AFM image of IGZO. (**e**) Schematic fabrication process of a-IGZO TFT/a-Si:H photodiode integration and image of a flexible 1T-1D structure image sensor on CPI film.

**Figure 2 nanomaterials-13-02886-f002:**
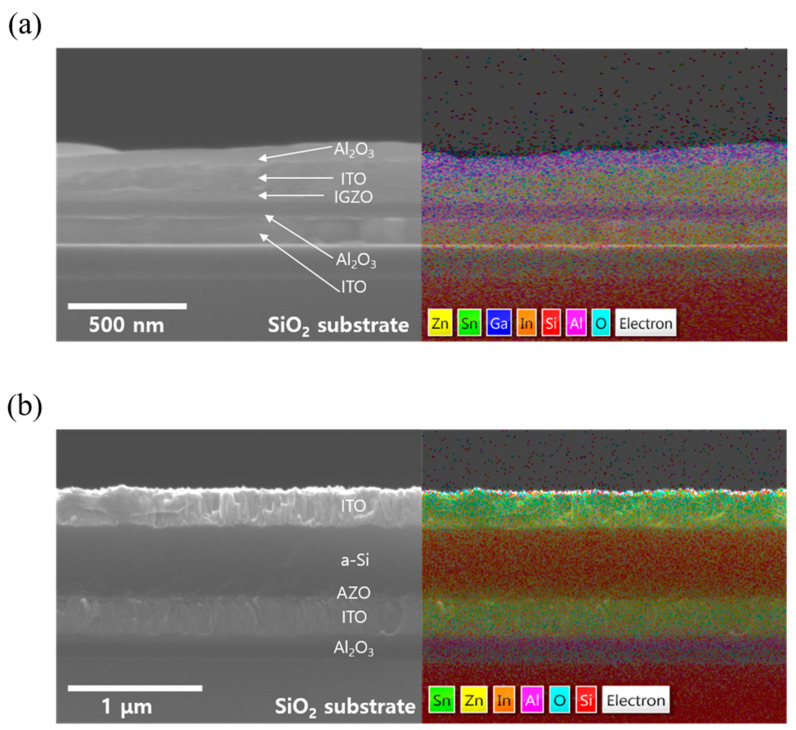
(**a**) SEM cross-section image and EDS map of the a-IGZO TFT. (**b**) SEM cross-section image and EDS map of the p-i-n a-Si:H photodiode.

**Figure 3 nanomaterials-13-02886-f003:**
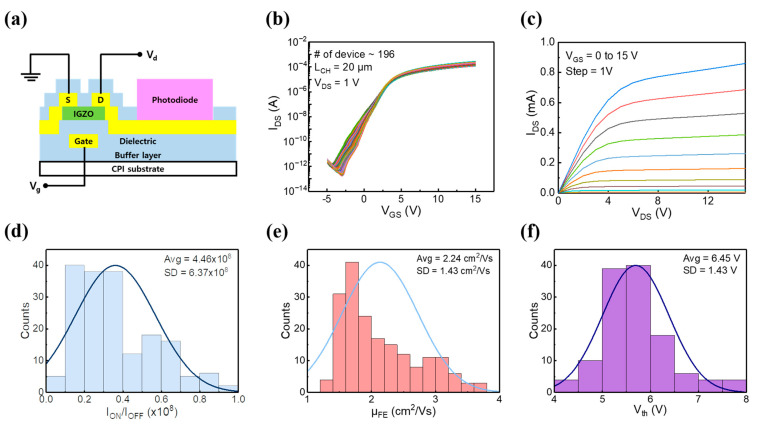
(**a**) Illustration of the transistor measurement method. (**b**) Transfer characteristics after 1T-1D integration when V_d_ is 1 V. (**c**) Output characteristic after integration when V_GS_ is 0–15 V. (**d**) On/off ratios, (**e**) field-effect mobilities, (**f**) and threshold voltages calculated for the characteristics of the 1T-1D integrated transistor.

**Figure 4 nanomaterials-13-02886-f004:**
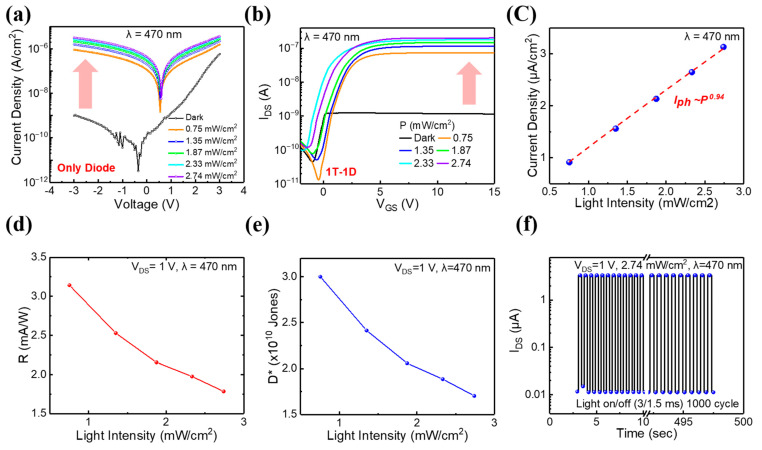
(**a**) J-V curve of the p-i-n a-Si:H photodiode under dark and light illumination at V_DS_—3 V (470 nm). (**b**) Transfer characteristics of a 1T-1D pixel exposed to various light intensities when V_DS_ is 1 V. (**c**) Photocurrent of p-i-n a-Si:H photodiode at various light intensities (470 nm). (**d**) Responsivity of p-i-n a-Si:H photodiode at various light intensities (470 nm). (**e**) Detectivity of p-i-n a-Si:H photodiode at various light intensities (470 nm). (**f**) Photoresponse under pulsed light illumination at 3 ms with 1000 cycles.

**Figure 5 nanomaterials-13-02886-f005:**
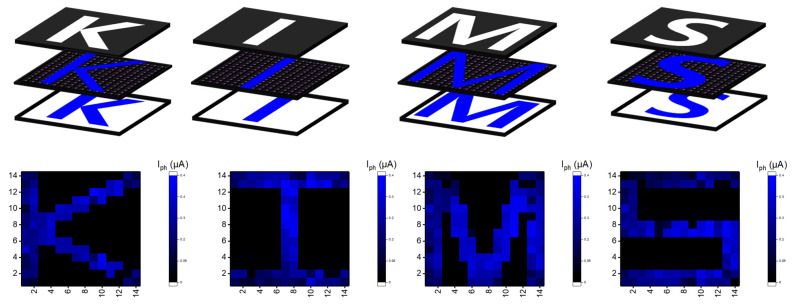
Schematic representation showing patterned I_ph_ mapping on a 14 × 14 active pixel image sensor array in a 1T-1D configuration under 2.74 mW/cm^2^ at a wavelength of 470 nm.

**Figure 6 nanomaterials-13-02886-f006:**
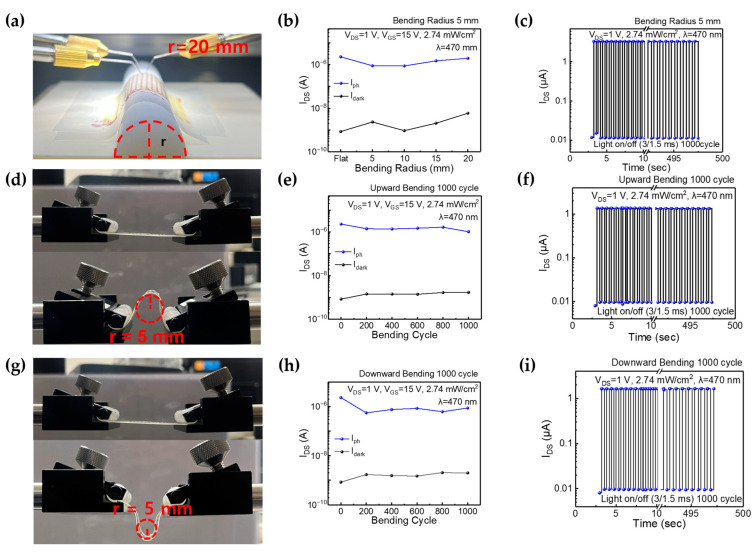
(**a**) When a bending radius of 20 mm was applied and measured, an image was obtained. (**b**) The difference in current between the dark and photo states of the 1T-1D integration transfer curve, with a bending radius of 5 mm applied (Vd = 1 V, Vg = 15 V). (**c**) Photoresponse under pulsed light illumination at 3 ms with 1000 cycles for a bending radius of 5 mm. (**d**) The image in the flat-upward bending radius in a 5 mm state. (**e**) The difference in current between the dark and photo states of the 1T-1D integration transfer curve after performing 1000 cycles of flat-upward bending with a radius of 5 mm. (**f**) Photoresponse under pulsed light illumination at 3 ms with 1000 cycles for a flat-upward bending radius of 5 mm after 1000 cycles. (**g**) The image is in a flat-downward bending radius 5 mm state. (**h**) The difference in current between the dark and photo states of the 1T-1D integration transfer curve after performing 1000 cycles of flat-downward bending with a radius of 5 mm. (**i**) Photoresponse under pulsed light illumination at 3 ms with 1000 cycles for a flat-downward bending radius of 5 mm after 1000 cycles.

**Table 1 nanomaterials-13-02886-t001:** Comparison of recently reported a-IGZO transistor/photodiode image sensors.

Device	Processing Temperature (°C)	Responsivity (mA/W)	Number of Array (EA)	Flexibility (Bending Radius/Bending Cycle)	Measurement Conditions	Reference
IGZO/a-Si:H	<90	31.43	14 × 14	>5 mm/>1000 cycle	λ = 470 nm	This work
IGZO/Perovskite/P3HT	<180	400	12 × 12	-	λ = 850 nm	[17]
IGZO/Perovskite/PCBM/BCP	-	340	12 × 12	-	λ = 550 nm	[18]
IGZO/Perovskite	-	200	12 × 12	-	λ = 530 nm	[43]

## Data Availability

The data that support the findings of this study are available from the corresponding author upon reasonable request.

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
