# Peer review of "Monolithic Integration of Semi-Transparent and Flexible Integrated Image Sensor Array with a-IGZO Thin-Film Transistors (TFTs) and p-i-n Hydrogenated Amorphous Silicon Photodiodes"

_nanomaterials, 2023, doi:10.3390/nano13212886_

Round 1
Reviewer 1 Report
Comments and Suggestions for Authors
Although this manuscript is of interest, major revision is recommended before acception with following concerns to address:
1. Considering both sides of the image sensor array are transparent, which side is used for light incidence in the imaging? In addition, the photoresponse-characterization of both sides should be supplied.
2. No light shield is applied in the IGZO TFT, is there any light-induced instability effect?
3. The SEM cross-section images of IGZO TFT and a-Si:H Photodiode are shown in Fig. S3, so why the SEM image of such monolithic integration structure is absent?
4. Is Fig. 3(a) the I-V curve of a-Si:H photodiode? There is no so-called "IDS-VDS curve" of a photodiode". If the data is obtained from the integrated device, please give a precise description and the corresponding measuring process.
5. Generally, a photodiode is operated at a reverse bias, but the authors state that the photodiode is under a voltage bias of 1 V in the full text, which is confusing.
6. The exponent in the power-law equation should be less than 1. Why the value is calculated to be 1.14 in Fig. 3 (c)?
7. What is the light intensity used for "KIMS" pattern imaging in Fig. 4?
8. Is the responsivity 31.43 mA/W of the photodiode? Why "responsivity exceeding 100 mA/W" is written in line 328? There are other inconsistencies in this manuscript, please check it carefully!
Comments on the Quality of English LanguageThe Egilish Language is fair.
Author Response
Once we have had sufficient time to carefully consider all of the reviewer’s concerns, we were able to further improve the quality of the revised manuscript. Thus, we provided and have made major changes as a full point-by-point response according to the reviewer’s suggestions as follows. This paper presents an important scientific breakthrough that should be of considerable interest to the broad readership of Nanomaterials.

Reviewer 2 Report
Comments and Suggestions for Authors
I would like to see SEM images of the devices, as well as optical microscopy and possibly AFM in the main paper. It seems reasonable for the authors to have such data already. With these additions, consideration for publication might proceed.
Author Response

(The authors gave the same response as above.)

Reviewer 3 Report
Comments and Suggestions for Authors
I appreciate the work performed by Choi et al. Please find my suggestion for improving the paper:
- There is a non-concordance between the sentence “Figure 1a illustrates the overall 3D scheme and scanning electron microscope (SEM) image of an individually a-IGZO TFT and a-Si:H Photodiode (1T-1D) image sensor, respectively” and the Figure 1 caption “Figure 1. a) Schematic of the a-IGZO TFT / a-Si:H Photodiode and Optical microscope image” so that is unclear if Fig 1a right is an optical image or a SEM image. However, in my opinion, the manuscript quality will be improved if the authors could add some top view microscopic images (SEM/AFM, etc) representative for the morphology of the deposited layers (either added to Fig 1 or as a separate figure, at the author’s best convenience). This is just a suggestion and not a mandatory request.
- The cross-sectional images from Figure S3 are of interest and, in my opinion, can be moved into the manuscript. As well, this is just a suggestion and not a mandatory request.
- Figure 2a as well does not contain any SEM image; but the authors stated that “The electrical circuit diagram of 1T-1D is shown in SEM image in Fig. 2a.” Fig 2.a caption suggests that there is only a scheme: “Illustration of the transistor measurement method”.
- Below the sentence “Additionally, the surface analysis of the a-IGZO channel material using X-ray Photoelectron Spectroscopy, as shown in Figure S4” please add some comments about the XPS results , particularly about the deconvolution of O1s peak (oxygen BEs).
Author Response

(The authors gave the same response as above.)

Reviewer 4 Report
Comments and Suggestions for Authors
A nice paper on flexible image sensors. In the introduction some more comments would be welcome on the impact of the technology chosen on the detector bandwidth and the ensuing application. e g. they develop a blue light detector, what is this for? how do they plan to extend the approach to other lambda ranges? The responsivity value reported in the comparison table with so many significant digit is much lower than the value reported in the conclusions.
Author Response

(The authors gave the same response as above.)

Round 2
Reviewer 1 Report
Comments and Suggestions for Authors
Now the authors have well addressed the issues raised by the reviewer and made corresponding piont by piont respone. I have no other comments on the manuscript.
Reviewer 2 Report
Comments and Suggestions for Authors
The revised version of the paper is improved and ready for publication